# ANGPTL2 promotes immune checkpoint inhibitor-related murine autoimmune myocarditis

Haruki Horiguchi [1,2], Tsuyoshi Kadomatsu [1,3✉], Tomoya Yamashita[1], Shinsei Yumoto[1,4], Kazutoyo Terada[1,3], Michio Sato [1], Jun Morinaga[1], Keishi Miyata [1,3] & Yuichi Oike [1,2,3✉]

Use of immune checkpoint inhibitors (ICIs) as cancer immunotherapy advances rapidly in the clinic. Despite their therapeutic benefits, ICIs can cause clinically significant immune-related adverse events (irAEs), including myocarditis. However, the cellular and molecular mechanisms regulating irAE remain unclear. Here, we investigate the function of Angiopoietin-like protein 2 (ANGPTL2), a potential inflammatory mediator, in a mouse model of ICI-related autoimmune myocarditis. ANGPTL2 deficiency attenuates autoimmune inflammation in these mice, an outcome associated with decreased numbers of T cells and macrophages. We also show that cardiac fibroblasts express abundant ANGPTL2. Importantly, cardiac myofibroblast-derived ANGPTL2 enhances expression of chemoattractants via the NF-κB pathway, accelerating T cell recruitment into heart tissues. Our findings suggest an immunostimulatory function for ANGPTL2 in the context of ICI-related autoimmune inflammation and highlight the pathophysiological significance of ANGPTL2-mediated cardiac myofibroblast/immune cell crosstalk in enhancing autoimmune responses. These findings overall provide insight into mechanisms regulating irAEs.

[1] Department of Molecular Genetics, Graduate School of Medical Science, Kumamoto University, Kumamoto 860-8556, Japan. [2] Department of Aging and Geriatric Medicine, Graduate School of Medical Science, Kumamoto University, Kumamoto 860-8556, Japan. [3] Center for Metabolic Regulation of Healthy Aging (CMHA), Graduate School of Medical Sciences, Kumamoto University, Kumamoto 860-8556, Japan. [4] Department of Gastroenterological Surgery, Graduate School of Medical Sciences, Kumamoto University, Kumamoto 860-8556, Japan. ✉email: tkado@gpo.kumamoto-u.ac.jp; oike@gpo.kumamoto-u.ac.jp

Cancer immunotherapy, with its recent focus on immune checkpoint inhibitors (ICIs), is a revolutionary approach that has shown great clinical benefit against multiple tumor types[1]. Nonetheless, ICIs can induce autoimmune side effects, called immune-related adverse events (irAEs), in almost any organ system[2]. irAEs occur in more than 70% of patients treated with nivolumab, which mediates blockade of programmed cell death protein 1 (PD-1)[3], or ipilimumab, which binds cytotoxic T-lymphocyte-associated protein 4 to potentiate T-cell responses[4]. One particularly problematic adverse effect following these treatments is life-threatening myocarditis. A study of data from eight clinical sites showed myocarditis prevalence to be 1.14% after ICI treatment, with a median time of onset of 34 days after starting ICIs[5]. However, a recent study suggested that cardiac irAEs may be underreported due to failure to diagnose asymptomatic cases[6]. Of note, the mortality rate of ICI-related myocarditis is reportedly as high as 50%[7]. It is generally accepted that mechanistically, ICIs can cause irAEs by enhancing the function of T cells targeting shared target antigen expressed on a cancer patient's tumor and heart cells, resulting in development of lethal autoimmune myocarditis[8–10]. Indeed, a recent study reported that cardiac and tumor infiltrates derived from melanoma patients who developed fatal myocarditis following ICI treatment shared high-frequency T cell receptor sequences[11]. Other potential predictors of irAEs are CD8+ T-cell clonal expansion[12] and high tumor mutational burden (TMB)[13]. However, the cellular and molecular mechanisms regulating irAE remain unclear. Defining mechanisms that underlie these adverse responses could suggest novel therapeutics to combat them.

Previously, we and others identified a family of secretory proteins structurally similar to angiopoietin and designated them angiopoietin-like proteins (ANGPTLs)[14,15]. We later reported that one of those proteins, ANGPTL2, contributes to maintenance of tissue homeostasis[16,17]. More recently, we showed that tumor stroma-derived ANGPTL2 enhances dendritic cell (DCs)-mediated CD8+ T cell anti-tumor immune responses[18]. We also reported that stroma-derived ANGPTL2 contributes to tumor suppression via macrophage-mediated CD4+ Th1 cell anti-tumor responses[19]. However, it remained unclear whether and how ANGPTL2-mediated immune responses contribute to irAEs.

Here, we report that cardiac fibroblast-derived ANGPTL2 plays a role in ICI-related autoimmune myocarditis by enhancing chemokine-induced recruitment of T cells. Our findings suggest that ANGPTL2-mediated crosstalk among cardiac stromal cells plays an essential role in irAE development.

## Results

**PD-1/PD-L1 blockade enhances progression of experimental autoimmune myocarditis.** During an antigen-specific anti-tumor immune response, neoantigens released from dead tumor cells are presented by DCs to T cells, and anti-tumor T cells could cross-react with the corresponding wild-type antigens in heart tissues[9]. As a potential mechanism regulating cardiac irAEs, ICIs enhance expansion and activation of T cells targeting an antigen shared by tumor and heart cells, resulting in damage to heart tissues[20]. Therefore, we established a murine model of experimental autoimmune myocarditis (EAM)[21] by adoptively transferring ex vivo-generated myosin heavy chain α (MyHCα)-presenting bone marrow-derived dendritic cells (BMDCs), which present a cardiac self-antigen to T cells in lymph nodes during sensitization, into WT mice on days 0, 2, and 4, and analyzed mice 10 days later (Fig. 1a). Control naive mice were not subjected to the EAM model or treated with ICIs. We observed scattered cardiac inflammation in hearts of EAM model mice but not in naive mice (Supplementary Fig. 1a). Histopathological analysis also revealed

extensive cardiomyocyte injury and an increased number of heart-infiltrating immune cells in EAM compared to naive mice (Fig. 1b, c). To assess ICI effects in this model, we simultaneously administered anti-PD-1 and anti-programmed death-ligand 1 (PD-L1) antibodies intraperitoneally into mice on days 1, 3, 5, 7, and 9 over the same 10-day period (Fig. 1a). Relative to untreated controls, ICI treatment extended inflammatory lesions (Supplementary Fig. 1a). ICI treatment also increased the area containing inflammatory cells in tissue sections relative to inflammatory areas seen in EAM-only samples (Fig. 1b, c) and enhanced CD45+ leukocyte infiltration of heart tissues (Fig. 1d). Histopathologic analysis also showed a substantial patchy lymphocytic infiltrate within the myocardium of mice subjected to EAM + ICI treatment (Fig. 1b), as seen in the myocardium of patients with ICI-related myocarditis[11]. These results suggest that ICI treatment exacerbates autoimmune inflammation in the EAM model.

Next, we performed flow cytometry analysis to characterize immune cell subpopulations infiltrating heart tissues of EAM model mice (Supplementary Fig. 1b). We found that, of immune cells analyzed, only the proportion of monocytes showed a significant decrease in hearts of EAM relative to naive mice (Supplementary Fig. 1c). In addition, quantification of immune cell distribution across cell clusters showed a relative increase in the number of T cells, DCs, and macrophages in hearts from EAM + ICI relative to EAM mice (Fig. 1e). We also observed a significant increase in the proportion of CD4+ and CD8+ T cells in hearts of EAM + ICI mice (Fig. 1f) and in absolute numbers of CD45+ total leukocytes, CD4+ T cells, CD8+ T cells, DCs, and macrophages infiltrating heart tissue (Fig. 1g). On the other hand, the number of monocytes infiltrating heart tissues was comparable in of EAM + ICI and EAM-only mice (Fig. 1g). Given that infiltration by T cells and macrophages is typically observed in the myocardium of patients with ICI-related myocarditis[11,22,23], we conclude that our ICI administration protocol in the EAM model mimics the pathology of ICI-related autoimmune myocarditis, and that T cells and s/macrophages are critical for the regulation of ICI-related autoimmune myocarditis.

Since myocarditis progresses to cardiomyopathy after inflammatory infiltrates disappear, we evaluated cardiac function in EAM and EAM + ICI mice 4 weeks after the initial DC immunization (Supplementary Fig. 2a). At that time point, inflammatory infiltrates had been resolved, and we observed interstitial fibrosis in hearts of EAM and EAM + ICI mice (Supplementary Fig. 2b, c). Echocardiography indicated that compared to EAM mice, EAM + ICI mice developed severe myocardial dysfunction, as determined by decreased fractional shortening (Fig. 1h, i) and decreased ejection fraction (Supplementary Fig. 2d). Furthermore, heart weight/body weight ratios significantly increased in EAM + ICI relative to EAM mice (Fig. 1j). These results overall suggest that in our experimental protocol, blocking PD-1/PD-L1 in an EAM model aggravates cardiac inflammation and accelerates cardiac dysfunction.

**Characterization of cardiac fibroblasts in EAM.** Cardiac fibroblasts play numerous roles in cardiac development and remodeling[24,25]. To characterize their function in the EAM model, we isolated CD45-CD31-PDGFRα+ cardiac fibroblasts from naive, EAM only, and EAM + ICI mice and analyzed their gene expression patterns (Fig. 2a). qRT-PCR analysis revealed that expression levels of *Pdgfra* and *Tcf21*, both markers of resident cardiac fibroblasts[26], in cardiac fibroblasts from the EAM group were markedly lower than those seen in cardiac fibroblasts of the naive group (Fig. 2b). By contrast, expression of *Acta2*, *Col1a1*, and *Col3a1*, all markers of myofibroblasts[26], in cardiac fibroblasts of the EAM group was significantly higher than that

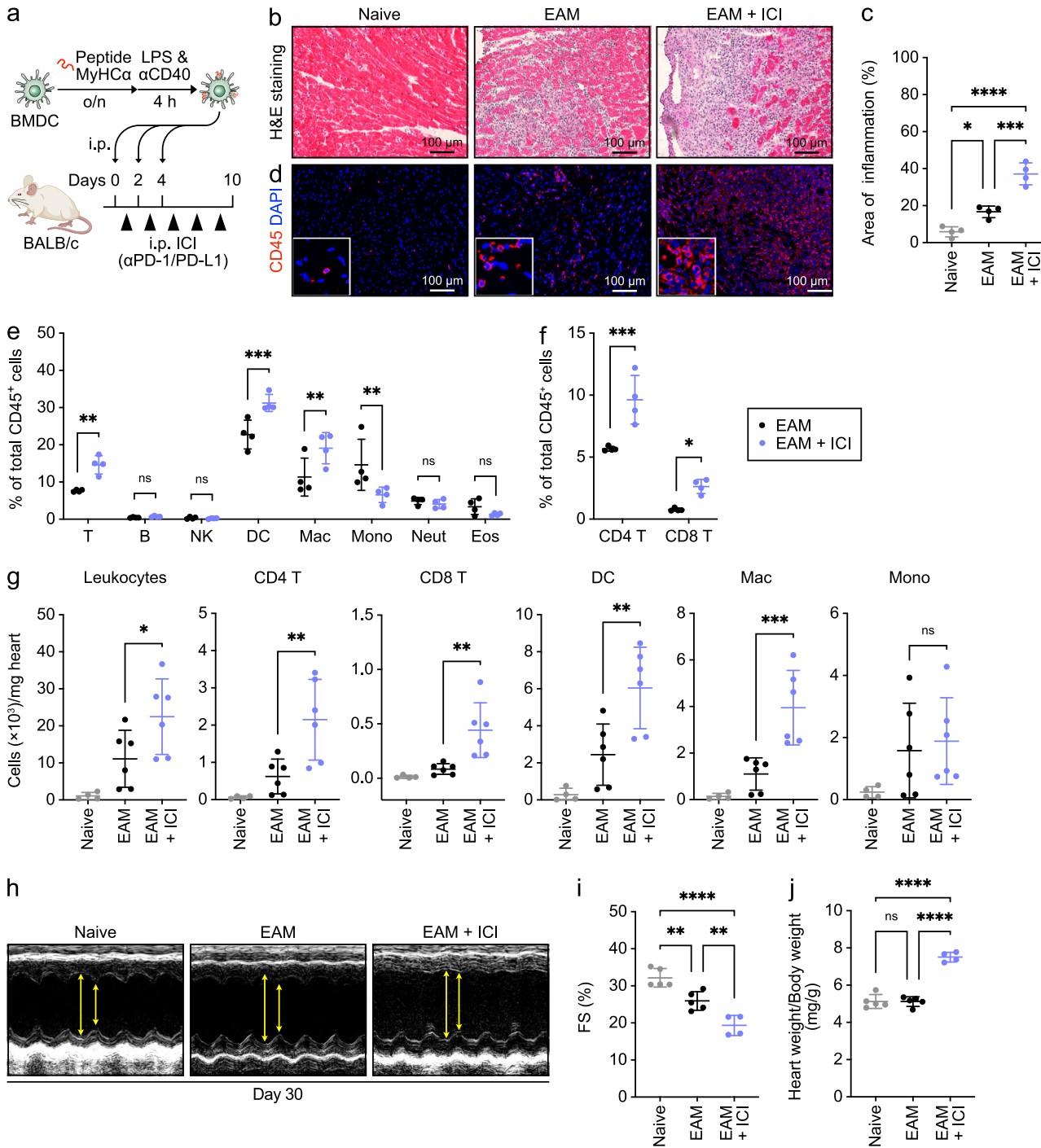

**Fig. 1 ICI administration exacerbates EAM. a** Schematic illustrating experimental design of the EAM model. **b** Representative images of H&E-stained heart tissues from indicated groups at day 10 of the EAM model. Scale bar, 100 μm. Naive mice were not subjected to the EAM model or treated with ICIs. **c** The area showing signs of inflammatory infiltration was scored from H&E-stained sections of ventricle shown in (**b**). Data are means ± SD; $n = 4$ per group. ****$P < 0.0001$; ***$P < 0.001$; *$P < 0.05$, one-way ANOVA test followed by Tukey's multiple comparison test. **d** Immunofluorescent staining for CD45 (red) in heart tissues in naive mice or on day 10 of the EAM or EAM + ICI model. Nuclei are counterstained with DAPI (blue). Scale bar, 100 μm. **e**, **f** Percentage of indicated leukocyte subsets infiltrating the heart among total CD45$^+$ cells at day 10 of the EAM or EAM + ICI model, based on flow cytometric analysis. Data are means ± SD; $n = 4$ per group. ns, not significant ($P > 0.05$); ***$P < 0.001$; **$P < 0.01$; *$P < 0.05$, two-way ANOVA test followed by Sidak's multiple comparison test. **g** Absolute number of indicated leukocyte subsets per milligram heart tissue on day 10 of indicated groups. Data are means ± SD; $n = 4$ for the naive group, $n = 6$ for the EAM group, and $n = 6$ for the EAM + ICI group. ns, not significant ($P > 0.05$); ***$P < 0.001$; **$P < 0.01$; *$P < 0.05$, one-way ANOVA test followed by Sidak's multiple comparison test. **h** Representative M-mode echocardiograms from indicated groups at day 30 of the EAM model. Arrows indicate distance between systolic contraction (LVESD) and diastolic relaxation (LVEDD). FS (%) (**i**) and Heart weight/Body weight ratios (**j**) in indicated groups at day 30 of the EAM model. Data are means ± SD; $n = 5$ for naive group, $n = 5$ for EAM group, and $n = 4$ for EAM + ICI group. ns, not significant ($P > 0.05$); ****$P < 0.0001$; **$P < 0.01$, one-way ANOVA test followed by Tukey's multiple comparison test.

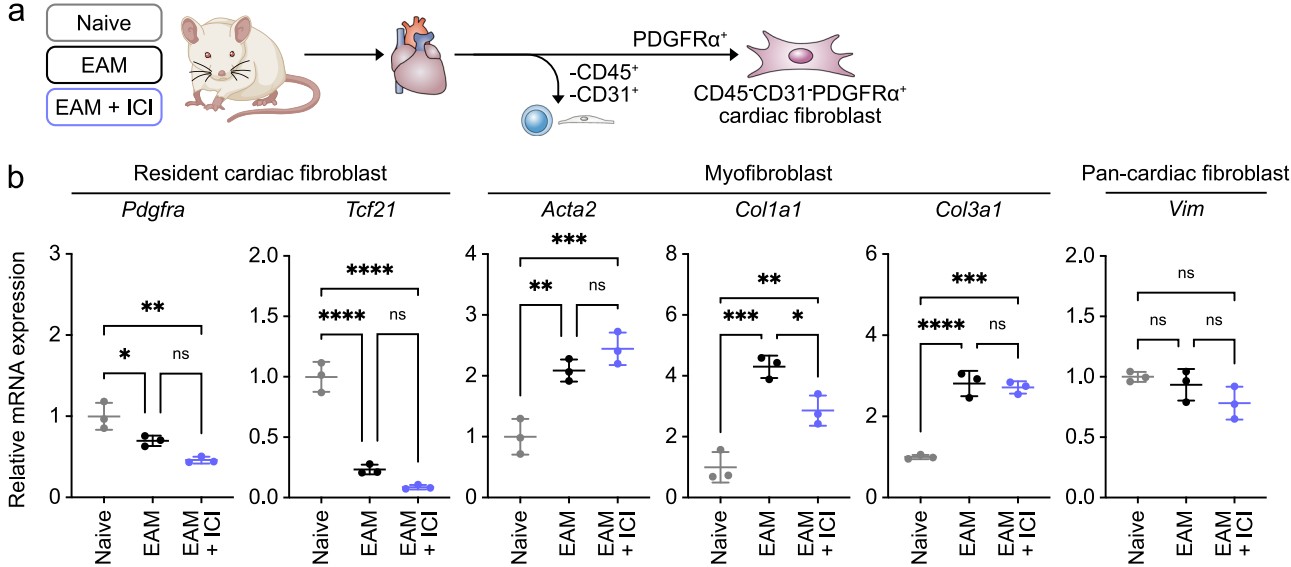

**Fig. 2 Phenotypic characterization of cardiac fibroblasts in the EAM. a** Schematic showing isolation of CD45⁻CD31⁻PDGFRα⁺ cardiac fibroblasts from naive, EAM only, and EAM + ICI mice. **b** qRT-PCR analysis of CD45⁻CD31⁻PDGFRα⁺ cardiac fibroblasts for transcripts of markers indicated at the top in indicated groups. Naive levels were set to 1. Data are means ± SD; $n = 3$ per group. ns, not significant ($P > 0.05$); ****$P < 0.0001$; ***$P < 0.001$; **$P < 0.01$; *$P < 0.05$, one-way ANOVA test followed by Tukey's multiple comparison test.

seen in naive cardiac fibroblasts (Fig. 2b), suggesting that cardiac fibroblasts acquire myofibroblast characteristics as EAM progresses. However, expression levels in cardiac fibroblasts of genes associated with general cardiac fibroblast identity, such as *Vim*, were comparable in EAM and naive groups (Fig. 2b). Also, when we compared cardiac fibroblasts in EAM and EAM + ICI groups, we did not observe significant differences in expression levels of *Pdgfra*, *Tcf21*, *Acta2*, *Col3a1*, and *Vim* (Fig. 2b), although *Col1a1* expression levels decreased in EAM + ICI group. Overall, these results suggest that in the EAM model, tissue-resident cardiac fibroblasts acquire myofibroblastic phenotypes, while ICI treatment does not have an effect on cardiac fibroblast conversion.

**Cardiac fibroblasts express abundant ANGPTL2.** Given that ANGPTL2 functions as an inflammatory mediator in various pathologic conditions[16,27], we asked whether ANGPTL2 contributes to irAE development. To do so, we first searched for the in vivo source of ANGPTL2 in heart tissues by assessing *Angptl2* transcript levels in immune and mesenchymal cells isolated from hearts of naive mice, such as bone marrow-derived macrophages (BMDMs), cultured T cells with T-cell receptor stimulation, and CD45⁻CD31⁻PDGFRα⁺ cardiac fibroblasts. qRT-PCR analysis revealed that cardiac fibroblasts express abundant *Angptl2* transcripts, while those transcripts were nearly undetectable in BMDMs and T cells (Fig. 3a). Immunohistochemical analysis of heart tissues from naive mice also revealed ANGPTL2 protein in mesenchymal stromal cells, especially in cardiac fibroblasts (Fig. 3b). At day 10, stromal cells also expressed ANGPTL2 protein in both the EAM and EAM + ICI models (Fig. 3b). Immunohistochemical analysis also demonstrated that cardiomyocytes express ANGPTL2 in the EAM + ICI model (Supplementary Fig. 3a). These results overall suggest that cardiac fibroblasts and/or some cardiomyocytes could be the in vivo source of ANGPTL2 in heart tissues. ANGPTL2 is a secreted protein, and its circulating levels increase with age and with chronic inflammatory diseases such as cancer, atherosclerosis, diabetes, and heart failure and with several age-related diseases[16,28]. Therefore, we assessed serum ANGPTL2 protein levels in our mouse model at time points up to 30 days (Supplementary Fig. 3b). In the EAM model, circulating ANGPTL2

levels modestly increased at day 10 (Supplementary Fig. 3c), although the change was not significant. By contrast, in the EAM + ICI model, we observed significantly increased serum ANGPTL2 levels at days 10 and 20 (Supplementary Fig. 3c), suggesting that ANGPTL2 activity is associated with ICI-related autoimmune myocarditis.

**ANGPTL2 deficiency decreases inflammatory infiltrates in ICI-related EAM.** To assess a potential ANGPTL2 function in autoimmune myocarditis, we subjected both *Angptl2*-deficient (*Angptl2⁻/⁻*) and WT mice to the EAM only model and assessed development of inflammation 10 days later (Supplementary Fig. 4a). Histopathologic analysis revealed comparable immune cell infiltration between genotypes on day 10 (Supplementary Fig. 4b, c). However, 4 weeks after the first DC immunization, echocardiography revealed myocardial dysfunction in WT mice, while *Angptl2⁻/⁻* mice showed less dysfunction, as indicated by higher fractional shortening and ejection fraction (Supplementary Fig. 4d–f). These results suggest that ANGPTL2 is dispensable for autoimmune responses in the EAM model, but that it likely contributes to cardiac dysfunction after inflammatory infiltrates disappear.

To assess a potential association between ANGPTL2 and ICI-related autoimmune myocarditis, we subjected both *Angptl2⁻/⁻* and WT mice to the previously described EAM + ICI model (Fig. 4a). Ten days after the first DC immunization, we observed decreased numbers of heart-infiltrating immune cells in *Angptl2⁻/⁻* relative to WT mice (Fig. 4b, c), a finding confirmed by staining with the pan-leukocyte marker CD45 (Fig. 4d). Flow cytometry analysis revealed a significant decrease in the number of heart-infiltrating total leukocytes, CD4⁺ T cells, CD8⁺ T cells, DCs, and macrophages in *Angptl2⁻/⁻* relative to WT mice, while monocyte numbers in heart tissues were comparable in both genotypes (Fig. 4e). By 4 weeks after the initial DC immunization, reduced cardiac function seen in WT mice was suppressed in *Angptl2⁻/⁻* mice (Fig. 4f, g and Supplementary Fig. 4g). At that time point, *Angptl2⁻/⁻* mice also showed a decreased heart weight/ body weight ratio relative to WT mice (Fig. 4h). These results suggest that ANGPTL2 contributes to autoimmune inflammation and chronic heart dysfunction seen in ICI-related myocarditis.

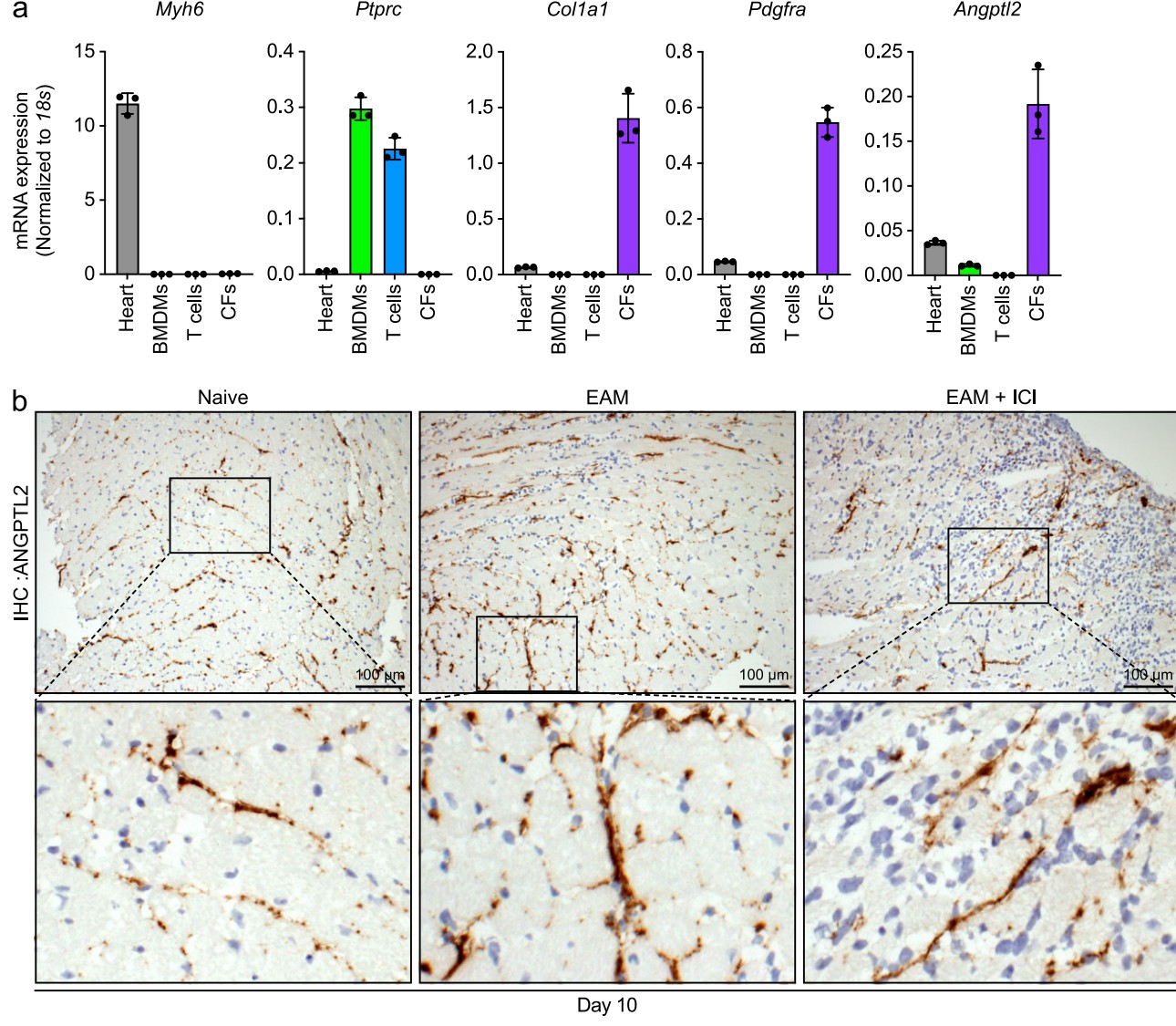

**Fig. 3 ANGPTL2 in heart is expressed predominantly by cardiac fibroblasts. a** mRNA levels of indicated genes in WT whole heart tissues, BMDMs, in vitro-stimulated T cells, or CD45⁻CD31⁻PDGFRα⁺ cardiac fibroblasts (CFs). Values are normalized to those of 18s RNA. **b** Representative images of ANGPTL2 immunostaining in heart tissues from indicated groups at day 10. Scale bar, 100 μm.

**ANGPTL2 expression in cardiac fibroblasts is required for chemokine-induced T cell recruitment**. We next addressed potential immunoregulatory mechanisms underlying ANGPTL2 activity in the context of ICI-related myocarditis. To investigate ANGPTL2 function in myofibroblasts, we used cultured myofibroblastic cardiac fibroblasts (myCFs) from mouse heart tissues of WT or *Angptl2*⁻/⁻ mice (Fig. 5a). Cultured WT myCFs were characterized by lower expression of *Pdgfra* and higher expression of *Acta2* than PDGFRα⁺ resident cardiac fibroblasts (Fig. 5b), and WT and *Angptl2*⁻/⁻ myCFs showed comparable *Pdgfra* and *Acta2* expression (Fig. 5c, d), suggesting that cardiac fibroblast-derived ANGPTL2 does not alter myofibroblastic characteristics of cardiac fibroblasts. Cardiac myofibroblasts modulate immune cell migration by producing chemokines[29]. Given that ANGPTL2 deficiency decreases the number of heart-infiltrating T cells in the EAM + ICI model (Fig. 4e), we hypothesized that cardiac myofibroblast-derived ANGPTL2 may regulate chemokine-induced infiltration of T cells. To test this hypothesis, we assessed chemokine expression in cultured WT and *Angptl2*⁻/⁻ myCFs. qRT-PCR analysis revealed significantly lower expression of T cell-attracting chemokines *Ccl3/4/5* and *Cxcl10/11* transcripts in

*Angptl2*⁻/⁻ relative to WT myCFs (Fig. 5e). To assess the impact of ANGPTL2-induced chemoattractant production on T cell recruitment, we cultured splenic T cells with conditioned medium from WT or *Angptl2*⁻/⁻ myCFs in Transwell chambers, with T cells cultured in the upper chamber and myCF-conditioned medium in the lower (Fig. 5f). That analysis revealed significantly reduced T cell migration activity in the presence of *Angptl2*⁻/⁻ versus WT myCF-conditioned medium (Fig. 5g), suggesting that chemokine production by ANGPTL2-expressing cardiac fibroblasts promotes T cell recruitment.

**ANGPTL2 upregulates chemokine expression via the NF-κB pathway**. To investigate whether ANGPTL2 derived from myCFs directly enhances chemokine expression, we treated cultured *Angptl2*⁻/⁻ myCFs with recombinant ANGPTL2 protein (rANGPTL2). Relative to untreated *Angptl2*⁻/⁻ cells, rANGPTL2-treated *Angptl2*⁻/⁻ myCFs showed significantly increased *Ccl3/5* and *Cxcl10/11* expression (Fig. 6a). Treatment of *Angptl2*⁻/⁻ CFs with exogenous ANGPTL2 also rescued chemokine expression to WT expression levels. (Supplementary Fig. 5) Furthermore, we cultured splenic T cells with conditioned medium from untreated

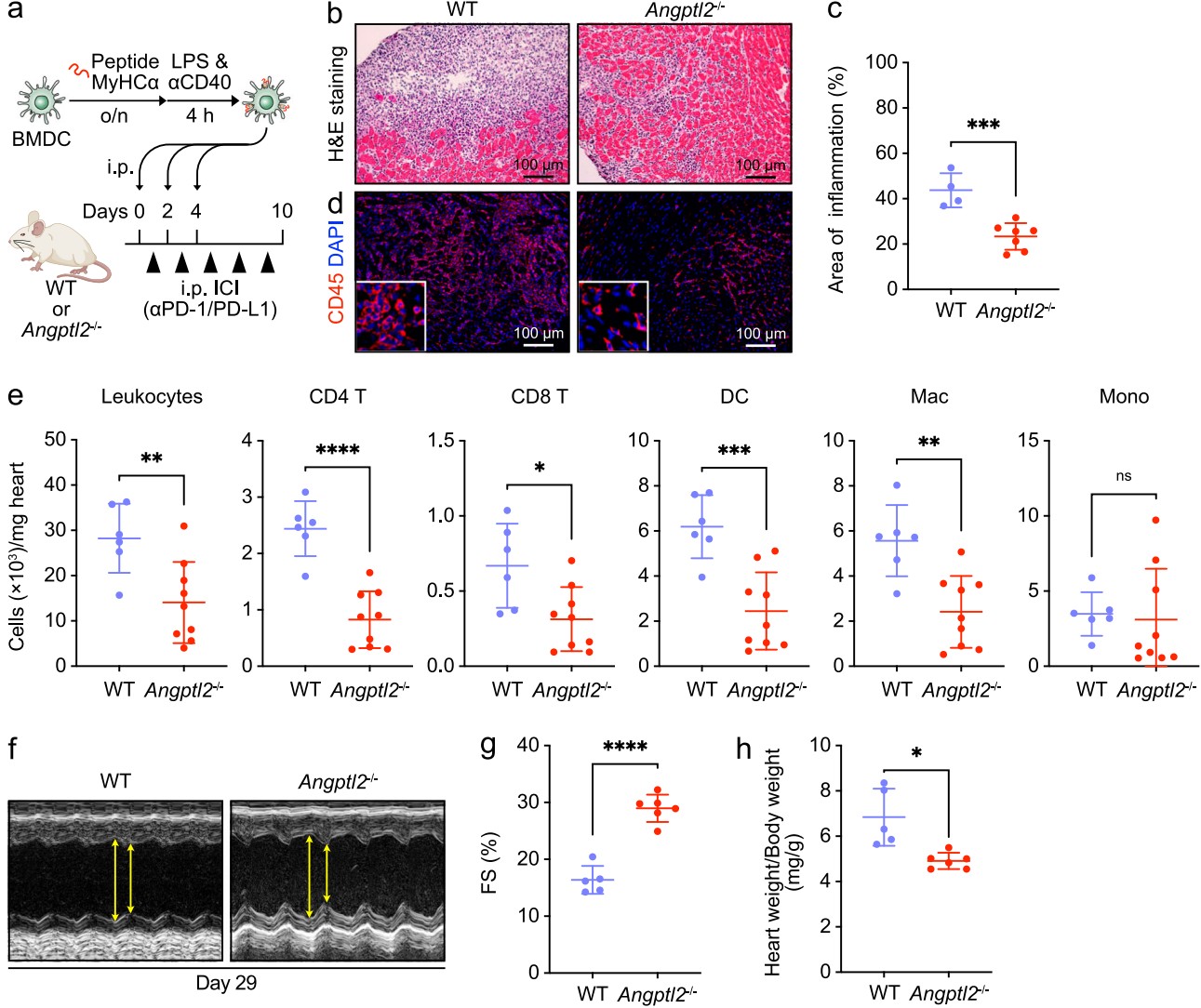

**Fig. 4 *Angptl2*-deficient mice show attenuated ICI-related EAM. a** Schematic illustrating experimental design of the EAM + ICI model. **b** Representative images of H&E-stained heart tissues from WT and *Angptl2*$^{-/-}$ mice at day 10 of the EAM + ICI model. Scale bar, 100 μm. **c** Areas of inflammatory infiltration were scored from H&E-stained sections of ventricle shown in (**b**). Data are means ± SD; n = 4 for WT group, n = 7 for *Angptl2*$^{-/-}$ group. ***P < 0.001, unpaired t test. **d** Immunofluorescent staining for CD45 (red) in heart tissues from indicated genotypes on day 10 of the EAM + ICI model. Nuclei are counterstained with DAPI (blue). Scale bar, 100 μm. **e** Absolute number of indicated leukocyte subsets per milligram heart tissue from WT and *Angptl2*$^{-/-}$ mice on day 10 of the EAM + ICI model. Data are means ± SD; n = 6 for WT group, n = 9 for *Angptl2*$^{-/-}$ group. ns, not significant (P > 0.05); ****P < 0.0001; ***P < 0.001; **P < 0.01; *P < 0.05, unpaired t test or Mann–Whitney test. **f** Representative M-mode echocardiograms from indicated groups at day 30 of the EAM + ICI model. Arrows indicate distance between systolic contraction (LVESD) and diastolic relaxation (LVEDD). FS (%) (**g**) and Heart weight/Body weight ratios (**h**) in indicated groups at day 28–29 of the EAM + ICI model. Data are means ± SD; n = 5 for WT, n = 6 for *Angptl2*$^{-/-}$ group. ****P < 0.0001; *P < 0.05, unpaired t test (**g**) or unpaired t test with Welch's correction (**h**).

*Angptl2*$^{-/-}$ cells or rANGPTL2-treated *Angptl2*$^{-/-}$ myCFs in Transwell chambers, with T cells cultured in the upper chamber and myCF-conditioned medium in the lower. Transwell analysis revealed significantly enhanced T cell migration activity in *Angptl2*$^{-/-}$ myCFs treated with rANGPTL2 (Fig. 6b), suggesting that ANGPTL2 expression by cardiac myofibroblasts enhances chemoattractant activity.

Finally, we investigated molecular mechanisms underlying chemokine upregulation by ANGPTL2. We previously reported that integrin α5β1 can serve as an ANGPTL2 receptor[27,30,31]. Therefore, we asked whether integrin α5β1 functions in ANGPTL2-dependent chemokine induction in myCFs. Flow cytometry analysis revealed that myCFs express integrin α5 and β1, but not another reported ANGPTL2 receptor, namely,

murine-paired immunoglobulin-like receptor-B [PIR-B; also known as LILRB2 (leukocyte immunoglobulin-like receptor B2) in humans][32] (Fig. 6c). The ANGPTL2–integrin α5β1 axis reportedly activates NF-κB signaling in endothelial cells[27], and intestinal fibroblasts[17]. Furthermore, NF-κB signaling regulates chemokine[33–35]. To determine whether ANGPTL2 regulates chemokine induction in myCFs via NF-κB signaling, we treated cultures of *Angptl2*$^{-/-}$ myCFs with the NF-κB inhibitor BAY11-7085 in the presence or absence of rANGPTL2 and then assayed chemokine transcript levels in those cells. BAY11-7085 treatment significantly attenuated rANGPTL2-induced chemokine expression in *Angptl2*$^{-/-}$ myCFs (Fig. 6d), suggesting overall that ANGPTL2 activates NF-κB signaling via integrin α5β1 to enhance chemokine production by cardiac myofibroblasts.

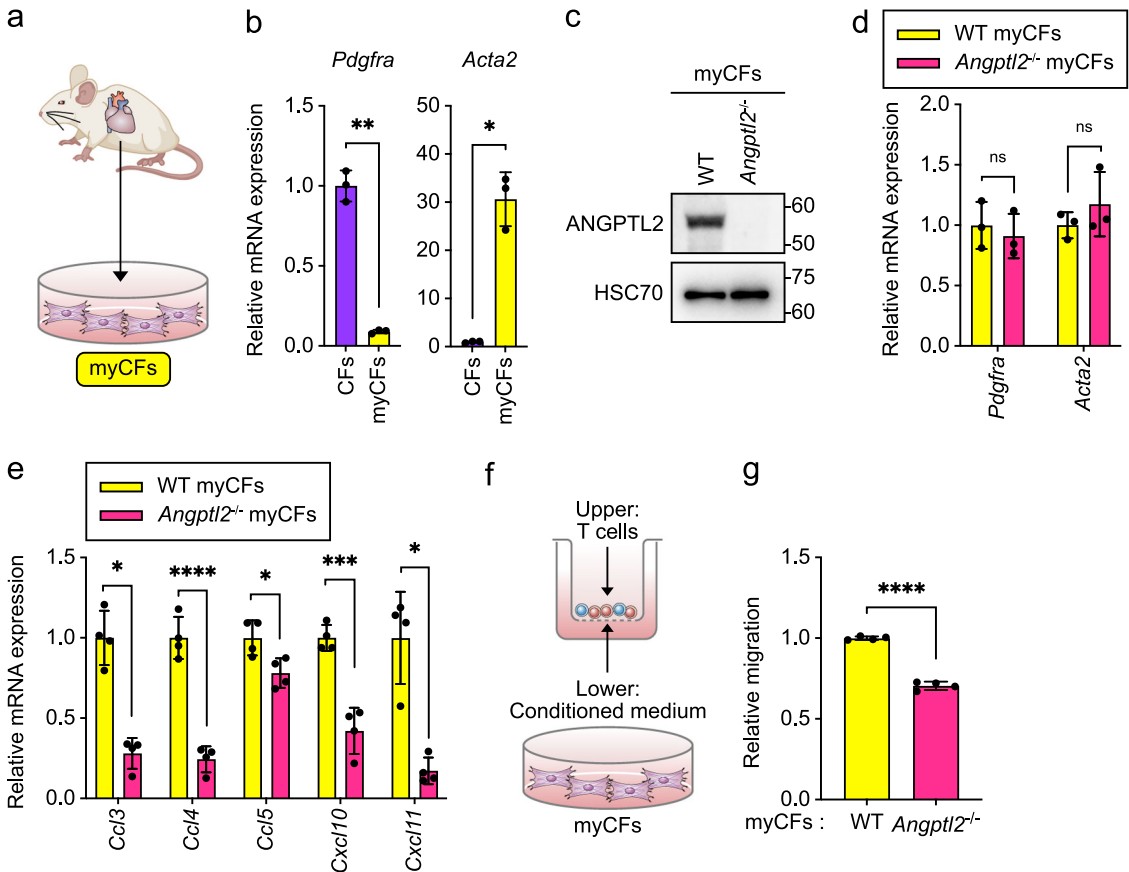

**Fig. 5 Chemokine expression is downregulated in *Angptl2⁻ᐟ⁻* myCFs. a** Myofibroblastic cardiac fibroblasts (myCFs) were isolated from heart and cultured. **b** *Pdgfra* and *Acta2* transcript levels in CD45⁻CD31⁻PDGFRα⁺ CFs or myCFs. CF levels were set to 1. Data are means ± SD; $n = 3$ per group. **$P < 0.01$; *$P < 0.05$, unpaired $t$ test with Welch's correction. **c** Representative immunoblotting for ANGPTL2 in WT and *Angptl2⁻ᐟ⁻* myCFs. HSC70 served as a loading control. **d** *Pdgfra* and *Acta2* transcript levels in WT or *Angptl2⁻ᐟ⁻* myCFs. WT myCF levels were set to 1. Data are means ± SD; $n = 3$ per group. ns, not significant ($P > 0.05$), unpaired $t$ test. **e** qRT-PCR analysis of transcripts of indicated genes in WT or *Angptl2⁻ᐟ⁻* myCFs. WT myCF levels were set to 1. Data are means ± SD; $n = 4$ per group. ****$P < 0.0001$; ***$P < 0.001$; *$P < 0.05$, unpaired $t$ test or Mann–Whitney test. **f** Schematic showing design of T cell migration assay. **g** Quantification of results from T cell migration assay of indicated groups based on flow cytometric analysis. WT myCF levels were set to 1. Data are means ± SD; $n = 4$ per group. ****$P < 0.0001$, unpaired $t$ test.

## Discussion

Here, we demonstrate that ANGPTL2 deficiency attenuates ICI-related autoimmune myocarditis, a condition mediated by T cells. Moreover, we present data suggesting that ANGPTL2 activates the NF-κB pathway in cardiac myofibroblasts to promote chemokine expression and recruitment of T cells. We conclude that cardiac myofibroblast-derived ANGPTL2 may enhance ICI-related autoimmune myocarditis (Fig. 7). Our data indicate that myofibroblast-derived chemokines, such as CCL3/4/5 and CXCL10/11, contribute to T cell recruitment to heart tissues, leading to irAE development.

Recognition of MyHCα by CD8⁺ T cells is necessary for ICI-related myocarditis[36]. Consistent with this view, we demonstrate here that ICI administration promotes CD8 T cell immune responses in an EAM model. ICI treatment also increased the number of CD4⁺ T cells and DCs/macrophages in heart tissues of mice with EAM. T cells activate monocyte-derived DC/macrophage inflammation in a way that facilitates T cell activation and proliferation, suggesting a T cell–DC/macrophage amplification loop in the EAM model[37]. Furthermore, others suggest that ICIs permit T cell activation to promote T cell and macrophage inflammation within heart tissue[38]. Our findings, combined with those of others, support the idea that CD8⁺ T cells are required to initiate cardiac irAE, and DC/macrophage inflammation occurs downstream of T cell responses. Moreover, we demonstrated that

ANGPTL2 deficiency decreased the appearance of key hallmarks of cardiac irAEs, suggesting that ANGPTL2 may be a critical driver of cardiac irAEs.

Cardiac fibroblasts play key roles in the structural and mechanical maintenance of the heart[29]. With injury, resident cardiac fibroblasts undergo conversion to myofibroblasts to mediate healing after acute myocardial infarction[39]. Furthermore, novel experimental approaches targeting cardiac myofibroblasts are promising potential therapies against heart disease[40]. The present study also demonstrates that resident cardiac fibroblasts acquire myofibroblast characteristics during autoimmune inflammation. Myofibroblasts modulate immune cell migration by producing chemokines[29,41]. Here, we demonstrated that cardiac myofibroblast-derived ANGPTL2 enhances *Ccl3/4/5* and *Cxcl10/11* transcript induction via the NF-κB pathway in an autocrine manner, thereby attracting T cells. We also previously reported that ANGPTL2–integrin α5β1 signaling activates the NF-κB pathway in intestinal myofibroblasts[17] and recently showed a significant decrease in the number of CD4⁺ T cells in colon lamina propria of *Angptl2⁻ᐟ⁻* relative to WT mice following injury[19]. These findings suggest that ANGPTL2-dependent chemoattractant activity contributes to immune responses in various organs.

A recent study analyzing ICI-related inflammatory arthritis revealed that effector CD8⁺ T cells recruited via CXCL9/10/11/16 signaling

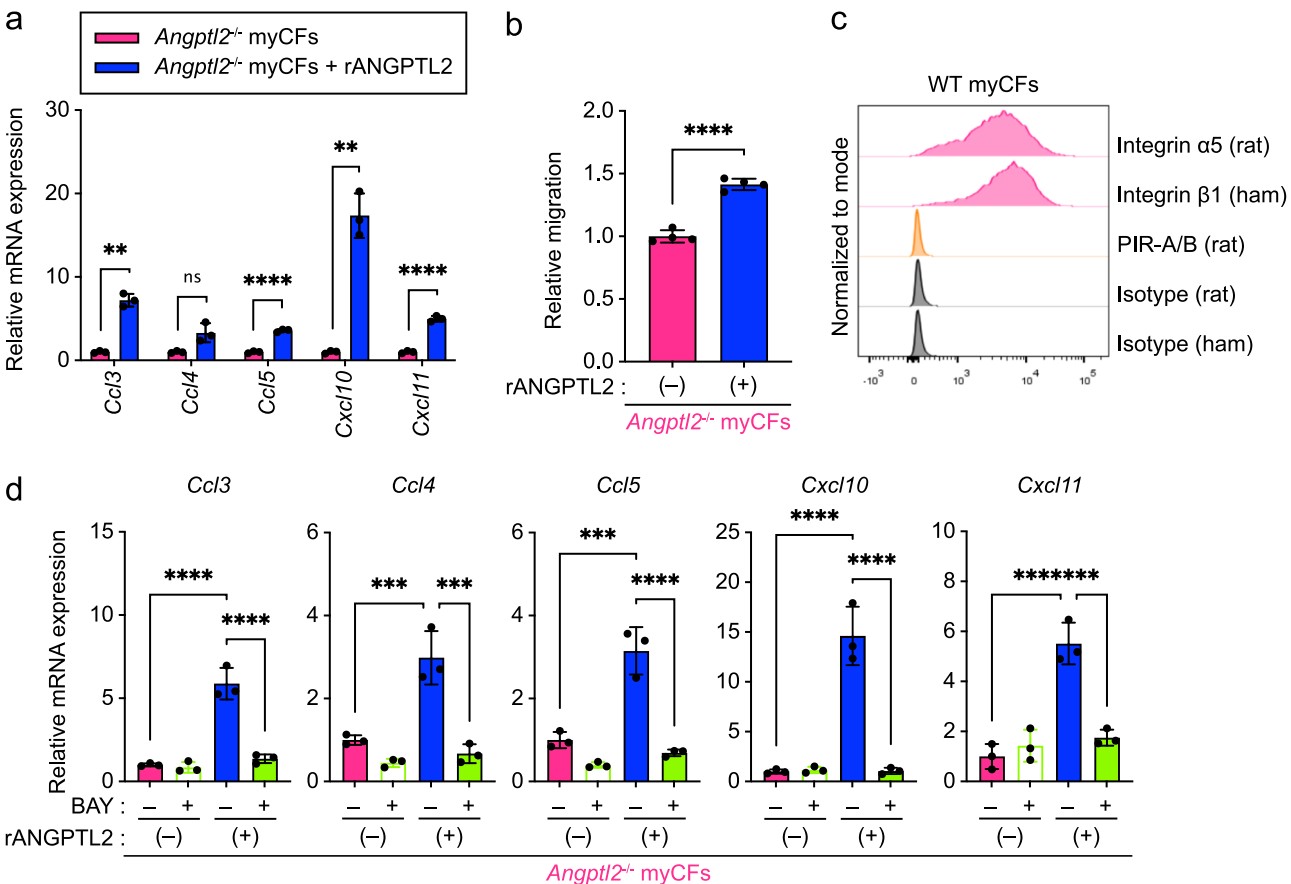

**Fig. 6 ANGPTL2 promotes chemokine expression in myCFs via NF-κB signaling. a** qRT-PCR analysis of transcripts of indicated genes in *Angptl2*$^{-/-}$ myCFs treated with or without rANGPTL2. Levels in ANGPTL2-untreated *Angptl2*$^{-/-}$ myCF were set to 1. Data are means ± SD; $n = 3$ per group. ns, not significant ($P > 0.05$); ****$P < 0.0001$; **$P < 0.01$, unpaired $t$ test or unpaired $t$ test with Welch's correction. **b** Quantification of T cell migration assay of indicated groups based on flow cytometric analysis. Levels in the presence of ANGPTL2-untreated *Angptl2*$^{-/-}$ myCF were set to 1. Data are means ± SD; $n = 4$ per group. ****$P < 0.0001$, unpaired $t$ test. **c** Representative histograms of cell surface expression of integrin α5, β1, or PIR-B in WT myCFs. **d** qRT-PCR analysis of transcripts of indicated genes in rANGPTL2-stimulated *Angptl2*$^{-/-}$ myCFs pretreated with the NF-κB inhibitor BAY11-7085 (BAY) or vehicle (DMSO). Expression levels in *Angptl2*$^{-/-}$ myCFs not treated with either rANGPTL2 or BAY were set to 1. Data are means ± SD; $n = 4$ per group. ****$P < 0.0001$; ***$P < 0.001$, one-way ANOVA test followed by Tukey's multiple comparison test.

play an important role in irAE[42]. Moreover, patients who developed various types of irAEs reportedly exhibited greater increases in serum CXCL9/10 levels post-ICI treatment than did patients without irAEs[43]. Others suggest that CCL4/5 chemokines may serve as attractive diagnostic and therapeutic targets in detecting and antagonizing life-threatening cardiac irAE in cancer patients treated with ICI[44]. These findings support the idea that ANGPTL2-mediated chemokine production functions in development of various irAEs. The relationship between ANGPTL2 expression and other irAEs, such as arthritis, dermatitis, pneumonitis, and colitis, warrants further investigation.

In this study, *Angptl2*-deficient mice showed a suppressed autoimmune inflammatory response compared to WT mice in the EAM + ICI model, effects not seen in the EAM model. One explanation of this difference could be that in the absence of ICI treatment, ANGPTL2 loss cannot dampen the immune response due to overriding activity of immune checkpoint molecules. Interestingly, *Angptl2*-deficient mice showed less myocardial dysfunction in the EAM model despite the fact that cardiac inflammation was comparable in WT and *Angptl2*-deficient mice. Moreover, surprisingly, cardiac function in *Angptl2*-deficient mice in the EAM + ICI model was similar to that seen in naive WT mice, although we observed cardiac inflammation in *Angptl2*-deficient mice of the EAM + ICI model. We previously reported that ANGPTL2 activity in cardiac pathologies

accelerates heart failure by perturbing cardiac function and energy metabolism; in that study, we demonstrated that pathologic stimuli, such as hypertension, increase cardiomyocyte ANGPTL2 expression, leading to a predisposition to heart failure[45]. Those findings suggest that ANGPTL2 expression in heart tissues including fibroblasts and cardiomyocytes may enhance susceptibility to cardiac dysfunction independent of ANGPTL2-mediated immune responses in the EAM and EAM + ICI model.

We and others have reported that circulating ANGPTL2 levels increase in various lifestyle- or aging-associated diseases[16,28], including cardiac dysfunction[46]. Here, we demonstrate that levels of circulating ANGPTL2 protein increased in a murine ICI-related autoimmune myocarditis model. Relevant to clinical application, serum ANGPTL2 concentration may be a useful parameter to predict irAE development. Currently used markers[47], such as troponin or N-terminal prohormone of brain natriuretic peptide (NT-pro BNP), can predict cardiac irAE development by detecting myocardial injury or cardiac dysfunction, respectively. Therefore, assessment of clinical features in combination with troponin, NT-pro BNP and ANGPTL2 during ICI treatment may facilitate diagnosis of ICI-associated cardiotoxicity. In addition, give our finding that ANGPTL2 activity underlies ICI-related autoimmune myocarditis, assessing a factor

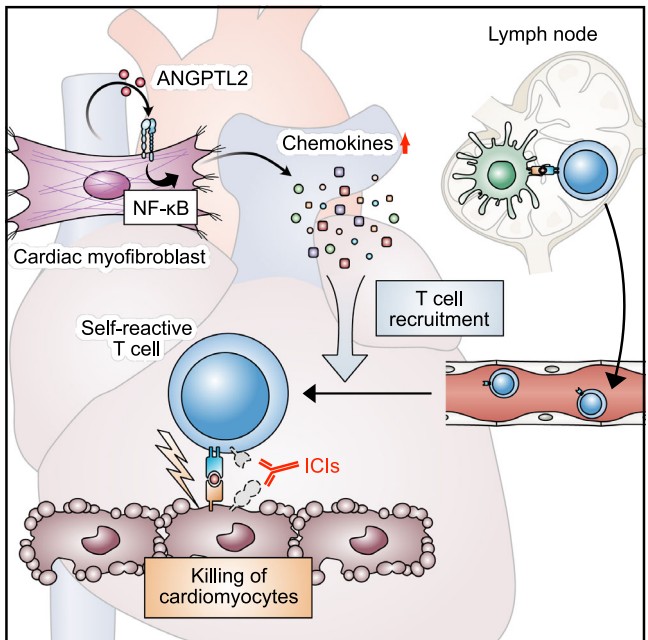

**Fig. 7 Model proposing regulation of ICI-related autoimmune myocarditis by ANGPTL2 derived from cardiac fibroblasts.** In heart tissues, cardiac myofibroblasts express abundant ANGPTL2, when then promotes chemokine expression in myofibroblasts via integrin α5β1 signaling, enhancing recruitment of T cells and contributing to autoimmune inflammation.

that serves as a causal factor of cardiotoxicity prior to ICI treatment could predict future development of irAE.

Elevated ANGPTL2 seen in the sera from the EAM + ICI model could be due to increased ANGPTL2 expression by cardiac fibroblasts. Alternatively, ANGPTL2 protein levels may increase in the bloodstream as ANGPTL2-expressing cardiomyocytes undergo destruction and lysis. Further studies are required to confirm these possibilities.

We recently demonstrated that cancer-associated fibroblast (CAF)-derived ANGPTL2 exhibits anti-tumor activities in an AOM/DSS mouse model of colitis-associated colon cancer, in a mouse kidney cancer model, and in murine syngeneic tumor models[18,19]. Mechanistically, we found that CAF-derived ANGPTL2 activates DCs and macrophages in a paracrine manner, facilitating anti-tumor immune responses by T cells. However, mechanisms underlying ANGPTL2-mediated tumor suppression are complex and not yet fully understood. It is generally accepted that a decline in production of chemokines such as CCL5 and CXCL9/10/11 in tumor stromal cells contributes to tumor promotion and even to ICI resistance[48–51]. Our present study suggests that an ANGPTL2–chemokine axis in CAFs may attract T cells to tumor tissues and function in tumor suppression, although further studies are needed to confirm this possibility.

In this study, we demonstrated that ANGPTL2–integrin α5β1 signaling in cardiac fibroblasts plays an immunostimulatory role in the context of autoimmune responses. In addition, we previously showed that ANGPTL2–integrin α5β1 signaling in tumor cells has tumor-promoting activity[30,31,52]. However, recently, we also showed that ANGPTL2–PIR-B signaling promotes DC activation and maturation and subsequent CD8+ T cell cross-priming, facilitating anti-tumor immune responses[18]. Thus, we speculate that blocking ANGPTL2–integrin α5β1 signaling could antagonize development of irAEs without significantly compromising anti-cancer immunity.

However, our study has some limitations. First, our DC-based EAM model mimics only the process after T cell sensitization by DCs and does not consider the association of irAEs with TMB or the steps of DC maturation. Given that ANGPTL2 facilitates DC maturation within tumor microenvironment[18], further studies are needed to determine whether ANGPTL2 expression in tumor microenvironment contributes to cardiac irAEs. Furthermore, in the DC-based EAM model, inflammation peaked 10 days after DC immunization[21], suggestive of acute immune responses. However, in clinical practice, myocarditis is reportedly diagnosed during the first 13 to 64 days of ICI treatment[11]. This temporal difference may be a species difference or possibly due to the fact that clinical data was derived from ICI-treated cancer patients, while mice assessed here did not harbor malignancies. Therefore, patients' cancer status may influence either the timing or extent of responses to ICI treatment.

In conclusion, our findings strongly suggest that ANGPTL2 promotes production of chemoattractants by cardiac myofibroblasts, which then recruits T cells into heart tissues. These findings illustrate the relevance of ANGPTL2-mediated communication between stroma and immune cells via chemokines to ICI-related autoimmune inflammation in heart tissue.

## Methods

**Animals**. BALB/c mice were purchased from CLEA (Japan). Angptl2+/- mice[27] were backcrossed to a BALB/c strain for 10 or more generations. 8–10-week-old male mice were used for all experiments, except for those shown in Supplementary Fig. 4d–f, which were performed using both male and female mice. The sample sizes were based on literatures. Data collection and analysis were conducted in a randomized and blinded manner. Mice showing clear signs of abnormal sickness were excluded from the study. All experimental procedures were approved by the Ethics Review Committee for Animal Experimentation of Kumamoto University.

**Induction of EAM**. BMDC-induced EAM was performed as described[21] with minor modifications. Briefly, BM cells were obtained from mouse femur and tibia and cultured in RPMI 1640 medium (Wako) containing 10% FCS, 100 units/ml penicillin, 100 μg/ml streptomycin, 50 μM 2-mercaptoethanol, and 20 ng/mL GM-CSF (BioLegend). After 3 days, an equal volume of complete medium containing 20 ng/mL GM-CSF was added. At day 6, non-adherent cells were collected and used as immature BMDCs. Immature DCs were pulsed overnight with 10 μg/ml of mouse MyHCα peptide (Ac-RSLKLMATLFSTYASADR-OH, AnaSpec). DCs were then activated for 4 h with 1 μg/ml LPS and 5 μg/ml of anti-CD40 antibody (BioLegend). $2 \times 10^5$ activated BMDCs (or PBS control) were intraperitoneally injected into recipient mice on days 0, 2, and 4. For ICI treatment studies, mice were treated with control or anti-PD-1 (BioXcell)/PD-L1 (BioXcell) antibodies (8 mg kg$^{-1}$) on days 1, 3, 5, 7, and 9.

**Quantitation of serum ANGPTL2 protein by ELISA**. Serum ANGPTL2 concentrations were measured by enzyme-linked immunosorbent assay (ELISA) using an ANGPTL2 Assay Kit (IBL), based on the manufacturer's instructions.

**Echocardiography**. Mice were preconditioned by chest hair removal with a topical depilatory (FujiFilm VisualSonics), anaesthetized with 1.5–2.5% isoflurane administered via inhalation, and maintained in a supine position on a dedicated animal handling platform with limbs attached for electrocardiogram gating during imaging. Body temperature was kept constant by feeding the signal of a rectal probe back to a heating pad, while

heart and respiratory rates were continuously monitored. Transthoracic echocardiography was performed using a high frequency ultrasound system dedicated to small animal imaging (VisualSonics Vevo 3100, FujiFilm VisualSonics) using a MX 400 linear array transducer (20–46 MHz). Experiments for Supplementary Fig. 4d–f were performed using a high frequency ultrasound system dedicated to small animal imaging (VisualSonics Vevo 2100, FujiFilm VisualSonics) and a MS 400 linear array transducer (18–38 MHz). M-mode recording was performed at the midventricular level. Percent LV fractional shortening (%FS) was calculated from M-mode measurements. All images were analyzed using dedicated software (Vevo Labo version 5.7.1 or Vevo 2100 version 1.4).

**Histology, immunohistochemistry, immunofluorescence**. For histological analysis, heart tissue samples were embedded in OCT compound, frozen in hexane with dry ice, sectioned with a cryostat, fixed in pre-cold acetone for 20 min and stained with hematoxylin and eosin (H&E). Cardiac inflammation was evaluated by histopathological microscopic approximation of the percent area of myocardium infiltrated with mononuclear cells using BZ-X analyzer version 1.4.1.1 (Keyence). For immunofluorescence, after fixation, samples were blocked with 10% serum for 20 min at room temperature and incubated with primary antibodies overnight at 4 °C, and then incubated with antibodies conjugated Alexa Flour 594 for 60 min at room temperature. Nuclei were stained with DAPI for 20 min. Anti-CD45 antibodies (1:100, BD, BD550539) were used. Immunohistochemical staining for mouse ANGPTL2 was performed as described[17]. For Masson's trichrome staining, heart tissue samples were fixed with 15% neutral buffered formalin, embedded in paraffin, sectioned with a microtome, and stained with Masson's Trichrome using routine procedures.

**Heart tissue digestion and cardiac fibroblast isolation**. The heart was minced into 2–3 mm$^3$ pieces and digested with 1 mg/ml collagenase D and 0.5 mg/ml dispase for 30 min on a shaking 37 °C incubator. Tissue was then passed through 100-µm cell strainer by mechanical disruption. Digested tissue was treated with red blood cell lysis buffer (0.15 M $NH_4Cl$, 10 mM $KHCO_3$, and 0.1 mM EDTA). The cell suspension was centrifuged and suspended in MACS buffer (Miltenyi Biotec). CD45$^-$CD31$^-$ PDGFRα$^+$ cardiac fibroblasts were isolated from digested heart specimens by MACS using a CD45, CD31, or CD140a MicroBead kit (Miltenyi Biotec).

**Flow cytometry**. Cells were suspended in MACS buffer and incubated with anti-CD16/32 mAb (BioLegend, 101310) for Fc receptor blocking. Cells were stained with the following fluorochrome-conjugated antibodies: PerCP/Cy5.5 anti-CD45 (BioLegend, 103131), APC anti-CD45 (BioLegend, 103111), PE/Cy7 anti-CD11b (Biolegend, 101215), FITC anti-CD3 (BD Biosciences, 553061), PE anti-CD4 (BD Biosciences, 553049), APC anti-CD8 (BioLegend, 100712), PerCP/Cy5.5 anti-CD11c (BioLegend, 117328), PE/Cy7 anti-MHC II (BioLegend, 107630), FITC anti-CD11b (BioLegend, 101206), V421 anti-CD64 (BioLegend, 139309), PE anti-Ly6G (BioLegend, 127607), APC anti-B220 (BioLegend, 103211), and PE anti-NK1.1 (BD Biosciences, 553165). Stained cells were analyzed by BD FACSVerse (BD bioscience). Data analysis was performed using FlowJo software (Treestar).

**myCF culture and treatment**. Digested heart tissues were cultured in RPMI-5 medium (RPMI with 5% FCS, 10 mM HEPES, 2 mM L-glutamine, 1 mM sodium pyruvate, penicillin/

streptomycin, and 2-mercaptoethanol). After 3 h incubation, non-adherent cells were washed away. One week later, only myofibroblasts remained after the first passage. myCFs were treated 6 h with vehicle or rANGPTL2 protein (10 µg/mL). For NF-κB inhibition, myCFs were pretreated with 5 µM BAY11-7085 (Sigma-Aldrich) for 120 min and then treated with rANGPTL2 in the presence of BAY11-7085.

**Isolation and culture of T cells**. Pan T cells were isolated from spleen of WT mice by MACS using a pan T cell isolation kit (Miltenyi Biotec), according to manufacturer's instruction. For T cell activation, cells were cultured 72 h in RPMI 1640 medium (Wako) containing 10% FCS, penicillin, streptomycin, 50 µM 2-mercaptoethanol, and Dynabeads mouse T-activator CD3/CD28 (Thermo Fisher Scientific).

**T cell migration assay**. myCF conditioned medium was collected from $1 \times 10^5$ myCFs cultured 2 days as monocultures. Then, T cells ($1 \times 10^6$) in 0.5% FCS-containing RPMI medium were cultured on a Transwell cell culture membrane (Corning) with myCF conditioned medium in the well below. After 6 h of incubation, T cells migrated into the lower well were counted using a BD Accuri flow cytometer (BD bioscience).

**Total RNA extraction and real-time quantitative RT-PCR**. Total RNA was isolated from cells and tissues using TRIzol regent (Invitrogen). DNase-treated RNA was reversed-transcribed with a PrimeScript RT regent Kit (Takara Bio). PCR products were analyzed using a Thermal Cycler Dice Real Time System (Takara Bio). PCR primer sequences (forward and reverse, respectively) were 5′-GGAGGTTGGACTGTCATCCAGAG-3′ and 5′-GCCTT GGTTCGTCAGCCAGTA-3′ for mouse *Angptl2*; 5′-GCTGGAA GATGAGTGCTCAGAG-3′ and 5′-CCAGCCATCTCCTCTGT-TAGGT-3′ for mouse *Myh6*; 5′-CTTCAGTGGTCCCATTGTGG TG-3′ and 5′-TCAGACACCTCTGTCGCCTTAG-3′ for mouse *Ptprc*; 5′-GACATGTTCAGCTTTGTGGACCTC-3′ and 5′-GGG ACCCTTAGGCCATTGTGTA-3′ for mouse *Col1a1*; 5′-GCAGT TGCCTTACGACTCCAGA-3′ and 5′-GGTTTGAGCATCTT-CACAGCCAC-3′ for mouse *Pdgfra*; 5′-CGTCCAGCTACATC GCTCACTT-3′ and 5′-CAGGTCATTCTCTGGTTTGCCG-3′ for mouse *Tcf21*; 5′-CTCTCTTCCAGCCATCTTTCAT-3′ and 5′-TA TAGGTGGTTTCGTGGATGC-3′ for mouse *Acta2*; 5′-GACCAA AAGGTGATGCTGGACAG-3′ and 5′-CAAGACCTCGTG CTCCAGTTAG-3′ for mouse *Col3a1*; 5′-AAAGCGTGGCTGC-CAAGAAC-3′ and 5′-GTGACTGCACCTGTCTCCGGTA-3′ for mouse *Vim*; 5′-ACTGCCTGCTGCTTCTCCTACA-3′ and 5′-AT GACACCTGGCTGGGAGCAAA-3′ for mouse *Ccl3*; 5′-ACCC TCCCACTTCCTGCTGTTT-3′ and 5′-CTGTCTGCCTCTTTTG GTCAGG-3′ for mouse *Ccl4*; 5′-CCTGCTGCTTTGCCTACCT CTC-3′ and 5′-ACACACTTGGCGGTTCCTTCGA-3′ for mouse *Ccl5*; 5′-ATCATCCCTGCGAGCCTATCCT-3′ and 5′-GACCTT TTTTGGCTAAACGCTTTC-3′ for mouse *Cxcl10*; 5′-CCGAG-TAACGGCTGCGACAAAG-3′ and 5′-CCTGCATTATGAGGC-GAGCTTG-3′ for mouse *Cxcl11*; 5′-TTCTGGCCAACGGTCT AGACAAC-3′ and 5′-CCAGTGGTCTTGGTGTGCTGA-3′ for mouse 18s. Relative transcript abundance was normalized to that of 18s mRNA.

**Immunoblot analysis**. Solubilized proteins were subjected to SDS-PAGE, and proteins were electro-transferred to nitrocellulose membranes. Immunoblotting was performed with antibodies against ANGPTL2 (1:1000, R&D Systems, BAF2084) or HSC70 (1:2000, Santa Cruz Biotechnology, #sc7298). Immunodetection was carried out using an ECL kit (GE Healthcare) according to the manufacturer's protocol.

**Statistics and reproducibility**. Statistical analyses were performed using GraphPad prism 9 software (GraphPad Software). Statistical parameters and methods are reported in respective figures and figure legends. Results with $p$-values < 0.05 were considered significant ($P < 0.05 = *$; $P < 0.01 = **$; $P < 0.001 = ***$; $P < 0.0001$). Comparisons between two groups were performed using an unpaired two-tailed $t$ test, an unpaired $t$ test with Welch's correction, or the Mann–Whitney $U$-test. Comparisons between three or more groups were performed using one-way ANOVA with Tukey's or Sidak's multiple comparison test. For comparisons with two or more independent variable factors, we used two-way ANOVA/mixed-effects analysis followed by Sidak's multiple comparison test. As necessary, normality distribution of the data was assessed using the Shapir-Wilk test for sample sizes $n < 4$. In vivo experiments were replicated at least two times with similar results or were pooled from at least two independent experiments. In vitro experiments were repeated at least two times with similar results.

**Reporting summary**. Further information on research design is available in the Nature Portfolio Reporting Summary linked to this article.

## Data availability

The uncropped/unedited western blot images are included in Supplementary Fig. 6. The Source data are provided in Supplementary Data 1. All other data are available from the corresponding author on reasonable request.

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

## Acknowledgements

We thank Kiyoka Tabu, Noriko Shirai, and Sayomi Iwaki for technical assistance. This work was supported by the Scientific Research Fund of the Ministry of Education, Culture, Sports, Science and Technology (MEXT) of Japan (grant 21K15508 to HH), the Takeda Science Foundation (HH), a Tasaki Memorial Research Grant for 2022 (HH), and a grant from the Center for Metabolic Regulation of Healthy Aging (CMHA) (HH).

## Author contributions

HH designed the study, performed and analyzed most experiments, and wrote the paper. TK designed and supervised the study and wrote the paper. TY and SY performed experiments. KT provided recombinant ANGPTL2 protein. MS assisted with echocardiography. JM assisted with statistical analysis. KM provided *Angptl2* mutant mice. YO coordinated, designed, and supervised the study and wrote the paper. All authors discussed the data and commented on the paper.

## Competing interests

The authors declare no competing interests.
