## [Peer Review File · Communications Biology]

ANGPTL2 promotes immune checkpoint inhibitor-related murine autoimmune myocarditisReviewers' comments:

Reviewer #1 (Remarks to the Author):

ANGPTL2 promotes immune checkpoint inhibitor-1 related autoimmune myocarditis by Horiguchi et al (Manuscript ID: COMMSBIO-23-1294-T; Communications Biology).

The authors have reported immunostimulatory role for Angiopoietin-like protein 2 (ANGPTL2) in the context of immune checkpoint inhibitors (ICI)-related autoimmune inflammation using mouse model of ICI-related experimental autoimmune myocarditis. Further, this report demonstrates that ANGPTL2 upregulates chemokine (Ccl3/5 and Cxcl10/11) expression via the NF- κ B pathway in cardiac myofibroblasts, which promotes in the recruitment of T cells into heart tissues. These findings provide mechanistic understanding of the ANGPTL2-mediated communication between stroma and immune cells via chemokines to ICI-related autoimmune inflammation in heart tissue. Overall, research findings from this study are of great significance to understand mechanisms regulating immune related adverse events induced by ICIs. Great work!

Minor comments:

1. Figures 5e & 5g and 6a & 6b; provide relative mRNA expression profile of chemokines, Ccl3/4/5 and Cxcl10/11 in WT myCFs vs. Angptl2^{-/-} myCFs, and Angptl2^{-/-} myCFs vs. Angptl2^{-/-} myCFs + rANGPTL2, respectively.

Treatment of recombinant ANGPTL2 protein in Angptl2^{-/-} myCFs seem to restore expression of above chemokines, could you provide details on the extent of this increase in comparison to WT myCFs (perhaps as a Supplemental figure comparing chemokine expression between WT myCFs and Angptl2^{-/-} myCFs + rANGPTL2)?

2. Typo, line 83: it reads as '... EAE mouse model', instead of EAM mouse model? Please verify.

Reviewer #2 (Remarks to the Author):

In this review by Horiguchi et al., authors tend to highlight the function of Angiopoietin-like protein 2 (ANGPTL2), in a mouse model of ICI-related autoimmune myocarditis. They report that cardiac fibroblast-derived ANGPTL2 plays a role in ICI-related autoimmune myocarditis and suggest the involvement of enhanced chemokine-induced recruitment of T cells.

Cardiac irAEs are caused by enhanced function of T cells, but the mechanisms underlying are still poorly understood. ANGPTL2 could participate in tumor suppression via T cell anti-tumor responses.

To unravel the mechanism involved, Horiguchi et al., established a mouse model of experimental autoimmune myocarditis. To do so, they adoptively transferred ex vivo-generated myosin heavy chain alpha presenting bone marrow-derived dendritic cells (BMDCs), which present a cardiac self-antigen to T cells in lymph nodes, into WT mice or Angptl2^{-/-} mice. They performed in vivo echocardiographic phenotyping, ex vivo histological phenotyping, as well as in vitro characterization and pathophysiological mechanisms discovery.

The manuscript is well written, well structured, and pleasant to read.

The main strength of the manuscript is the use of in vivo and in vitro models, that gave rise to a real mechanistic hypothesis linking ICI-related autoimmune myocarditis, myofibroblasts, ANGPTL2, and NF κ B.

The main weakness is the low number of animals per condition.

The introduction could provide the reader with more information to understand the need for or the nature of the work.

The first paragraph of the introduction might not be necessary, too general and can be applied to any field of health research.

Giving more epidemiological data in the second paragraph would enhance the importance of this work: rate of irAEs, and specifically cardiac irAEs.

Results:

1) I. 76: Could you explain more clearly if the "naive" mice were injected with BMDC or not and/or ICI.

2) Figure 1: could you give an explanation regarding the heterogeneity of number of animals in the different groups and different assays?

3) Figure 1: ANOVA one-way test has been used to compare 3 animals per group, however this type of data would require a non-parametric test.

4) I.131: it appears that Angptl2 is expressed mainly in CFs, however Angptl2 is a circulatory protein, and plasma angptl2 levels increase with age and with various chronic inflammatory diseases such as cancer, atherosclerosis, diabetes, heart failure and a multitude of age-related diseases (10.3390/ijms222212232). This would need to be addressed in the discussion.

5) The IHC images Figure 3b do not provide sufficient resolution for the reader to assess the accuracy of ANGPTL2 expression in cardiac fibroblasts especially, as reported I.129.

Methods:

6) induction of EAM. Please provide concentration of antibiotics in culture medium. What age and sex were the mice used? Could you specify what control treatment was used? Have you controlled the biodistribution of the dose of BMDC used?

7) The model developed recapitulated an acute auto-immune event, with the appearance of a strong cardiac phenotype as shown Figure 1b. The H&E staining shows that CD45+ cells are present in non-collagenic tissue deposition. Commenting on this histological phenotype and linking its limit to the clinical relevance would be of interest.

8) Could you please specify the number of animals used?

Conclusion:

9) The paragraph proposing a working model (I.202) might be too speculative compared to the results highlighted in the manuscript.

Reviewer #3 (Remarks to the Author):

This study demonstrates that cardiac myofibroblast-derived ANGPTL2 contributes to ICI-related autoimmune myocarditis, in a murine model of experimental autoimmune myocarditis, by promoting the production of chemoattractants that, in turn, recruit T cells into cardiac tissue. The study is well conducted, the manuscript is clear and well written.

The major limitation of this study, notified by the authors themselves, is that this experimental model is not performed in a context of cancer. Another limitation is the potential implication for clinical practice: does this study implies that to prevent cardiac immune-related adverse events, ANGPTL2 circulating levels should be monitored? How better is ANGPTL2 when compared to the gold standard cardiac biomarkers NT-proBNP, or Troponin T?

Major comments:

1-The mouse model of experimental autoimmune myocarditis is complex and should be described in more details, in the methods. In addition: sex and age of the mice must be included; explain why control mice are not injected (with a control solution) despite the fact that EAM mice are subjected to 3 i.p injections followed by 5 ICI i.p injections; what is the rationale to study inflammation and gene expression 10 days after ICI initiation? Does this represent an acute response to ICI? In cancer patients treated with ICI, it has been reported that CV-irAEs occur within the first 6 weeks of ICI treatment.

Figure 1c shows area of inflammation from heart sections; please indicate the region (ventricle?) where the analysis was performed. Is the inflammatory response restricted to a particular zone, or generalized throughout the whole heart?

Figure 1e: why naïve mice are not presented, as in figure 1g?

Figure 1h and 1i: did the authors measure the inflammatory response at 4 weeks? Did it "disappear" as stated in line 94? In addition, other echo parameters (LVEF, values of LVESD, LVEDD, etc), but also heart rate or any signs of arrhythmias should be included to better characterize cardiac dysfunction in this model.

2-Figure 3a shows that cardiac fibroblasts from naïve mice express high levels of Angptl2. What is the expression of this gene in cardiomyocytes?

Does the CFs expression of Angptl2 increase in EAM and even further in EAM+ICI mice, as suggested by the IHC image in Figure 3b? In Figure 3b, at what time (Day 10? Day 30?) were the IHC performed? It would be interesting to measure Angptl2 expression (gene and/or protein) at different time points, to better understand the time course of this inflammatory marker during EAM and after ICI treatment.

3- Line 121-122: "Given that ANGPTL2 functions as an inflammatory mediator (...) we asked whether it contributes to irAE development". On the other hand, a non-inflammatory function of ANGPTL2 was reported by this team to contribute to cardiac dysfunction, in cardiomyocytes (Tian et al., 2016). Please reconcile and discuss.

4- Figure 4: What does EAM alone do in Angptl2^{-/-} mice? If available, please give more echo data or cardiac function data in Figure 4f-g-h.

When comparing data from Figure 1i and Figure 4g, it appears that FS (%) is very similar between naïve mice and Angptl2^{-/-}-EAM-ICI mice; does this mean that the absence of ANGPTL2 fully reversed the cardiac dysfunction induced by ICI? please discuss this.

5- Line 49: "Defining mechanisms that underlie these adverse responses could suggest novel therapeutics to combat them". Although this is theoretically true, what would be the approach to treat

ANGPTL2-related cardiac irAEs? In other words, how could ANGPTL2 be inhibited, as stated in the discussion, line 263-264, "blocking ANGPTL2-integrin $\alpha 5\beta 1$ signaling could antagonize the development of irAEs"?

Minor comments:

Line 43: "Cardiac irAEs are relatively rare." A recent study suggested that the incidence rate of cardiac irAEs is likely underdiagnosed, due to asymptomatic events (Isawa T et al., 2022, PMID: 35348766). Please, give more details concerning the rate of irAEs, and what are the known risk factors.

Define all acronyms, such as anti-PD1, anti-PD-L1 (line 77), TCR stimulation (line 125) or DC immunization (line 137).

Line 115-116: please correct "we did not observe significant differences in expression levels of (...), Col3a1, ...". According to Figure 2b, expression of Col3a1 is significantly decreased in EAM+ICI compared to EAM.

Line 146: Correct "chronic heart failure" by chronic heart dysfunction. There is no evidence of HF.

Line 182: "one reported ANGPTL2 receptor is integrin $\alpha 5\beta 1$ "; since all the references quoted are from this team, the sentence could be changed to "we reported".

Responses to comments by Reviewer 1

We thank Reviewer 1 very much for their evaluation of our manuscript.

Reviewer #1 (Remarks to the Author):

ANGPTL2 promotes immune checkpoint inhibitor-1 related autoimmune myocarditis by Horiguchi et al (Manuscript ID: COMMSBIO-23-1294-T; Communications Biology).

The authors have reported immunostimulatory role for Angiopoietin-like protein 2 (ANGPTL2) in the context of immune checkpoint inhibitors (ICI)-related autoimmune inflammation using mouse model of ICI-related experimental autoimmune myocarditis. Further, this report demonstrates that ANGPTL2 upregulates chemokine (Ccl3/5 and Cxcl10/11) expression via the NF- κ B pathway in cardiac myofibroblasts, which promotes in the recruitment of T cells into heart tissues. These findings provide mechanistic understanding of the ANGPTL2-mediated communication between stroma and immune cells via chemokines to ICI-related autoimmune inflammation in heart tissue. Overall, research findings from this study are of great significance to understand mechanisms regulating immune related adverse events induced by ICIs. Great work!

Minor comments:

1. Figures 5e & 5g and 6a & 6b; provide relative mRNA expression profile of chemokines, Ccl3/4/5 and Cxcl10/11 in WT myCFs vs. Angptl2^{-/-} myCFs, and Angptl2^{-/-} myCFs vs. Angptl2^{-/-} myCFs + rANGPTL2, respectively.

Treatment of recombinant ANGPTL2 protein in Angptl2^{-/-} myCFs seem to restore expression of above chemokines, could you provide details on the extent of this increase in comparison to WT myCFs (perhaps as a Supplemental figure comparing chemokine expression between WT myCFs and Angptl2^{-/-} myCFs + rANGPTL2)?

Thank you for these comments. Accordingly, we assessed chemokine expression by comparing WT myCFs with *Angptl2* KO myCFs treated with rANGPTL2. rANGPTL2 treatment rescued chemokine expression by *Angptl2* KO CFs to WT levels. (Supplementary Fig 5). We now report these findings in the Results on Page 11, Line 205–207.

2. Typo, line 83: it reads as ‘... EAE mouse model’, instead of EAM mouse model? Please verify.

We apologize for this mistake and have corrected it in the revision.

Responses to comments by Reviewer 2

We thank Reviewer 2 for constructive comments, which were extremely helpful. We have extensively revised our manuscript based on those suggestions and feel that it is much improved. Our answers are as follows.

Reviewer #2 (Remarks to the Author):

In this review by Horiguchi et al., authors tend to highlight the function of Angiotensin-like protein 2 (ANGPTL2), in a mouse model of ICI-related autoimmune myocarditis. They report that cardiac fibroblast-derived ANGPTL2 plays a role in ICI-related autoimmune myocarditis and suggest the involvement of enhanced chemokine-induced recruitment of T cells.

Cardiac irAEs are caused by enhanced function of T cells, but the mechanisms underlying are still poorly understood. ANGPTL2 could participate in tumor suppression via T cell anti-tumor responses.

To unravel the mechanism involved, Horiguchi et al., established a mouse model of experimental autoimmune myocarditis. To do so, they adoptively transferred ex vivo-generated myosin heavy chain alpha presenting bone marrow-derived dendritic cells (BMDCs), which present a cardiac self-antigen to T cells in lymph nodes, into WT mice or Angptl2^{-/-} mice. They performed in vivo echocardiographic phenotyping, ex vivo histological phenotyping, as well as in vitro characterization and pathophysiological mechanisms discovery.

The manuscript is well written, well structured, and pleasant to read.

The main strength of the manuscript is the use of in vivo and in vitro models, that gave rise to a real mechanistic hypothesis linking ICI-related autoimmune myocarditis, myofibroblasts, ANGPTL2, and NFκB.

The main weakness is the low number of animals per condition.

The introduction could provide the reader with more information to understand the need for or the nature of the work.

The first paragraph of the introduction might not be necessary, too general and can be applied to any field of health research.

Giving more epidemiological data in the second paragraph would enhance the importance of this work: rate of irAEs, and specifically cardiac irAEs.

Thank you for these suggestions. We have now omitted the first paragraph of the Introduction and added description relevant to irAEs to the Introduction on Page 3, Line 32–50.

Results:

1) l. 76: Could you explain more clearly if the “naive” mice were injected with BMDC or not and/or ICI.

Naive mice were *not* administered either BMDCs or ICIs, but rather treated with PBS and control IgG (as controls for BMDC injection and ICIs, respectively). We now state this clearly in the Results, Page 5, Line 74–75, and in the Methods, Page 18, Line 343–354.

2) Figure 1: could you give an explanation regarding the heterogeneity of number of animals in the different groups and different assays?

As noted, the number of mice in each group and in each assay differed in large part because we can breed only a limited number of mice for this project, making it challenging to use the same number of mice in each experiment. Thus we collected data independently from available mice for each experiment, and those numbers varied.

3) *Figure 1: ANOVA one-way test has been used to compare 3 animals per group, however this type of data would require a non-parametric test.*

We appreciate the question but note that in original Figure 1, we did not compare 3 animals. Thus, we feel the reviewer may be mistaken in citing Figure 1, and for the revision we made no changes in Figure 1. However, it is possible that the reviewer instead meant Figure 2, which *does* indeed show a comparison of three animals per group. Therefore, assuming this to be the case, we now address the reviewer's concern about appropriate statistical analysis and have now performed the Shapiro-Wilk test to analyze data shown in Figure 2. All of that data were found to be normally distributed. Therefore, we chose a one-way ANOVA test followed by Tukey's multiple comparison test. We now state this in the Methods section, Page 24, Line 473–474.

4) *l.131: it appears that Angptl2 is expressed mainly in CFs, however Angptl2 is a circulatory protein, and plasma angptl2 levels increase with age and with various chronic inflammatory diseases such as cancer, atherosclerosis, diabetes, heart failure and a multitude of age-related diseases (10.3390/ijms222212232). This would need to be addressed in the discussion.*

Based on Reviewer 3 suggestions, we assessed ANGPTL2 protein levels in sera at various time points in both the EAM and EAM + ICI models and observed a significant increase in serum ANGPTL2 levels in the EAM + ICI model only (Supplementary Fig. 3c). As noted, circulating ANGPTL2 levels increase in various lifestyle- or aging-associated diseases, including cardiac dysfunction, suggesting that ANGPTL2 could serve as a biomarker for development or progression of these conditions, and as a predictor of irAE development. We now report these findings in the Results on Page 8, Line 145–152, and in the Discussion on Page 15, Line 285–295.

5) *The IHC images Figure 3b do not provide sufficient resolution for the reader to assess the accuracy of ANGPTL2 expression in cardiac fibroblasts especially, as reported l.129.*

Accordingly we have added high resolution and higher magnification images to Figure 3b.

Methods:

6) *induction of EAM. Please provide concentration of antibiotics in culture medium. What age and sex were the mice used? Could you specify what control treatment was used? Have you controlled the biodistribution of the dose of BMDC used?*

BM cells were cultured in RPMI 1640 containing 10% FCS, 100 units/ml penicillin, 100 µg/ml streptomycin, 50 µM 2-mercaptoethanol, and 20 ng/mL GM-CSF. We employed 8–10-week-old male mice for experiments. Moreover, PBS and control IgG served as controls for BMDC injection and ICIs, respectively. We now state these points in the Methods on Page 18, Line 343–354.

We agree that controlling BMDC cell biodistribution could increase our model's reliability. However, we do not yet have a means to track BMDCs injected into mice and thus cannot make that assessment at this time. However, it is reported that i.p.-injected BMDCs accumulate in heart and mediastinal lymph nodes during EAM (Van der Borghet et al., PMID: 30524444).

7) *The model developed recapitulated an acute auto-immune event, with the appearance of a strong cardiac phenotype as shown Figure 1b. The H&E staining shows that CD45+ cells are present in non-collagenic tissue deposition. Commenting on this histological phenotype and linking its limit to the clinical relevance would be of interest.*

As you note, our histopathologic analysis shows a substantial patchy lymphocytic infiltrate within the myocardium, an outcome reported in the myocardium of patients with ICI-related myocarditis (Johnson et al., PMID: 27806233). We now state this in the Results section, Page 5, Line 84, through Page 6, Line 86, and cite the Johnson et al study.

8) *Could you please specify the number of animals used?*

Mouse numbers were as follows: WT BALB/c mice (n = 75) for Figure 1–3 and Supplementary Fig. 1–3; and *Angptl2*^{+/+} (WT) and *Angptl2*^{-/-} littermate mice (n = 30 and 39, respectively) for Figure 4–6 and Supplementary Fig. 4–5. Also, for BMDC generation, 10 or more WT mice were used. We now report these numbers in relevant figure legends.

Conclusion:

9) *The paragraph proposing a working model (l.202) might be too speculative compared to the results highlighted in the manuscript.*

Accordingly, we have now omitted the indicated paragraph (l.202 in the original Discussion section) and modified the Discussion (Page 13, Line 233–235) to decrease speculation.

Responses to comments by Reviewer 3

Thank you for your constructive comments, which were extremely helpful. We have extensively revised our manuscript based on those suggestions and feel that it is much improved. Our answers are as follows.

Reviewer #3 (Remarks to the Author):

This study demonstrates that cardiac myofibroblast-derived ANGPTL2 contributes to ICI-related autoimmune myocarditis, in a murine model of experimental autoimmune myocarditis, by promoting the production of chemoattractants that, in turn, recruit T cells into cardiac tissue. The study is well conducted, the manuscript is clear and well written.

The major limitation of this study, notified by the authors themselves, is that this experimental model is not performed in a context of cancer. Another limitation is the potential implication for clinical practice: does this study implies that to prevent cardiac immune-related adverse events, ANGPTL2 circulating levels should be monitored? How better is ANGPTL2 when compared to the gold standard cardiac biomarkers NT-proBNP, or Troponin T?

We now observed significantly increased serum ANGPTL2 levels in the EAM + ICI model (Supplementary Fig. 3c), suggesting that monitoring ANGPTL2 levels could predict irAE development. Currently, the troponin and NT-pro BNP markers predict cardiac irAE that has already developed by detecting myocardial injury and cardiac dysfunction, respectively. We feel that assessment of clinical features using a combination of troponin, NT-pro BNP and ANGPTL2 during ICI treatment could strengthen diagnosis of ICI-associated cardiotoxicity. Also, given our finding that ANGPTL2 activity underlies ICI-related autoimmune myocarditis, assessing a factor that serves as a causal factor of cardiotoxicity prior to ICI treatment could predict future development of irAE. We now report these findings in the Results on Page 8, Line 145–152, and in the Discussion on Page 15, Line 285–295.

Major comments:

1-The mouse model of experimental autoimmune myocarditis is complex and should be described in more details, in the methods. In addition:

sex and age of the mice must be included;

explain why control mice are not injected (with a control solution) despite the fact that EAM mice are subjected to 3 i.p injections followed by 5 ICI i.p injections;

what is the rationale to study inflammation and gene expression 10 days after ICI initiation? Does this represent an acute response to ICI? In cancer patients treated with ICI, it has been reported that CV-irAEs occur within the first 6 weeks of ICI treatment.

In response to these comments, we added text to the Methods (Page 18, Line 343–354) describing our murine model. In brief, it states that 8–10-week-old male mice were used for experiments and that PBS and control IgG served as respective controls for BMDC injection and ICIs.

In the DC-based EAM mouse model, inflammation peaks 10 days after DC immunization (Eriksson et al., PMID: 14625544), suggesting an acute immune response. However, as you and others (Johnson et al., PMID: 27806233) note, myocarditis is reportedly diagnosed 13-64 days after ICI treatment. This temporal difference may be due to species differences or possibly to the

fact that the clinical data was derived from ICI-treated cancer patients, while mice assessed here did not harbor malignancies. Therefore, patients' cancer status may influence either the timing or extent of responses to ICI treatment. We now state this in the Discussion on Page 17, Line 323–328.

Figure 1c shows area of inflammation from heart sections; please indicate the region (ventricle?) where the analysis was performed. Is the inflammatory response restricted to a particular zone, or generalized throughout the whole heart?

First, this panel shows analysis of ventricle stained with H&E, as now stated in the figure legend. Also, we observed that cardiac inflammation was scattered throughout heart of EAM model mice, rather than restricted to a particular area. In addition, we found that ICI treatment further extends inflammatory lesions (Supplementary Fig. 1a). We now report these findings in the Results on Page 5, Line 75–84.

Figure 1e: why naïve mice are not presented, as in figure 1g?

In the original study we did not include naïve mice in Fig. 1e because in that analysis we were focused on effects of ICI in the EAM model. However, to address the reviewer's concern, we have now performed flow cytometry analysis to characterize immune cell subpopulations infiltrating heart tissues of naïve compared to EAM only model mice. Those results, shown in new Supplementary Fig. 1c, indicate that among immune cell populations, only levels of monocytes changed (in this case, decreased) in hearts of EAM relative to naïve mice. We now report these findings in the Results on Page 6, Line 89–91.

Figure 1h and 1i: did the authors measure the inflammatory response at 4 weeks? Did it “disappear” as stated in line 94? In addition, other echo parameters (LVEF, values of LVESD, LVEDD, etc), but also heart rate or any signs of arrhythmias should be included to better characterize cardiac dysfunction in this model.

First, at week 4, the inflammatory infiltrates were resolved, and we observed interstitial fibrosis in hearts of EAM and EAM + ICI mice (Supplementary Fig. 2b, c). We conclude that at this point, inflammatory responses have disappeared and report these findings in the Results on Page 6, Line 104–106.

Second, we added echo parameters such as heart rate, end systolic diameter, end diastolic diameter, and ejection fraction (Supplementary Fig. 2d) and confirmed that EAM + ICI mice developed severe myocardial dysfunction, based on decreased ejection fraction. We now report these findings in the Results on Page 6, Line 106–108.

2-Figure 3a shows that cardiac fibroblasts from naïve mice express high levels of Angptl2. What is the expression of this gene in cardiomyocytes?

Does the CFs expression of Angptl2 increase in EAM and even further in EAM+ICI mice, as suggested by the IHC image in Figure 3b? In Figure 3b, at what time (Day 10? Day 30?) were the IHC performed? It would be interesting to measure Angptl2 expression (gene and/or protein) at different time points, to better understand the time course of this inflammatory marker during EAM and after ICI treatment.

First, we do not have access to cardiomyocyte-labeled (e.g. alpha MHC-EGFP Tg) mice, making it technically challenging to assay *Angptl2* transcripts in isolated cardiomyocytes. Thus, we assessed ANGPTL2 protein expression in heart tissue using immunohistochemistry from WT mice or mice subjected to the EAM + ICI model. As shown in Fig. 3b, at day 10 cardiac stromal cells express abundant ANGPTL2 in the EAM + ICI model. For the resubmission we subjected samples to longer DAB exposure and observed that cardiomyocytes from WT EAM + ICI mice also express ANGPTL2 protein (Supplementary Fig. 3a). Note that these signals were absent in similarly treated ANGPTL2 KO mice. We now report these findings in the Results on Page 8, Line 140–143.

Second, to monitor ANGPTL2 expression at different time points we assessed serum ANGPTL2 protein levels by ELISA at time points up to 30 days. In the EAM model, circulating ANGPTL2 increased modestly but not significantly at day 10 (Supplementary Fig. 3c), while in the EAM + ICI model, serum ANGPTL2 levels significantly increased at days 10 and 20 (Supplementary Fig. 3c), suggesting that serum ANGPTL2 levels are predictive of irAE development. Elevated ANGPTL2 in serum is likely due to increased ANGPTL2 expression in cardiac fibroblasts but could be due to ANGPTL2 release into the bloodstream from damaged cardiomyocytes. Further studies are required to confirm these possibilities. We now report these findings in the Results on Page 8, Line 145–152, and in the Discussion on Page 16, Line 296–299.

3- Line 121-122: “Given that ANGPTL2 functions as an inflammatory mediator (...) we asked whether it contributes to irAE development”. On the other hand, a non-inflammatory function of ANGPTL2 was reported by this team to contribute to cardiac dysfunction, in cardiomyocytes (Tian et al., 2016). Please reconcile and discuss.

4- Figure 4: What does EAM alone do in *Angptl2*^{-/-} mice? If available, please give more echo data or cardiac function data in Figure 4f-g-h.

When comparing data from Figure 1i and Figure 4g, it appears that FS (%) is very similar between naïve mice and *Angptl2*^{-/-}-EAM-ICI mice; does this mean that the absence of ANGPTL2 fully reversed the cardiac dysfunction induced by ICI? please discuss this.

Regarding 3- and 4-, when we subjected both WT and *Angptl2* KO mice to the EAM model, immune cell infiltration into heart was comparable between genotypes (Supplementary Fig. 4b, c), possibly because although increased ANGPTL2 may evoke an autoimmune response, potential anti-inflammatory effects of ANGPTL2 deficiency in an EAM model may be overridden by activity of immune checkpoint molecules. Nonetheless, *Angptl2* KO mice showed less myocardial dysfunction in the EAM model (Supplementary Fig. 4d–f). Furthermore, as you note, cardiac function in naive WT and *Angptl2* KO mice in the EAM + ICI model is similar, despite the fact that the latter exhibit cardiac inflammation.

As for our previous report (Tian et al., PMID: 27677409), we found that ANGPTL2 activity in cardiac pathologies accelerates heart failure by perturbing cardiac function and energy metabolism (Tian et al., PMID: 27677409). These findings suggest that ANGPTL2 expression in heart tissues, including fibroblasts and cardiomyocytes, enhances susceptibility to cardiac dysfunction independent of ANGPTL2-mediated immune responses in the EAM and EAM + ICI models. We now report these findings in the Results on Page 9, Line 155–163, and in the Discussion on Page 14, Line 270, through Page 15, Line 284.

Finally, we have now added echo parameters to the analysis shown in original Fig. 4, such as heart rate, end systolic diameter, end diastolic diameter, and ejection fraction (see Supplementary Fig. 4g). We confirmed that *Angptl2* KO mice with EAM + ICI developed less myocardial dysfunction, based on increased ejection fraction. We now report these findings in the Results on Page 9, Line 171–173.

5- Line 49: “Defining mechanisms that underlie these adverse responses could suggest novel therapeutics to combat them”. Although this is theoretically true, what would be the approach to treat ANGPTL2-related cardiac irAEs? In other words, how could ANGPTL2 be inhibited, as stated in the discussion, line 263-264, “blocking ANGPTL2-integrin $\alpha 5\beta 1$ signaling could antagonize the development of irAEs”?

To begin to address this possibility, we are currently developing a neutralizing antibody to block interaction between ANGPTL2 and integrin $\alpha 5\beta 1$ and have several promising clones. In the future, we will assess whether these antibodies could reduce cardiac irAE development in an EAM + ICI mouse model, as a first step to translating these findings into human studies.

Minor comments:

Line 43: “Cardiac irAEs are relatively rare.” A recent study suggested that the incidence rate of cardiac irAEs is likely underdiagnosed, due to asymptomatic events (Isawa T et al., 2022, PMID: 35348766). Please, give more details concerning the rate of irAEs, and what are the known risk factors.

A study of data from eight clinical sites reported myocarditis prevalence after starting ICI to be 1.14% with a median time of onset of 34 days (Mahmood et al., PMID: 29567210). However, as you note, cardiac irAEs may have been underdiagnosed and underreported. Also, potential predictors of irAEs are TCR diversity (Johnson et al., PMID: 27806233), CD8+ T-cell clonal expansion (Subudhi et al., PMID: 27698113), and the TMB (Bomze et al., PMID: 31436791). We now state this in the Introduction on Page 3, Line 32–50.

Define all acronyms, such as anti-PD1, anti-PD-L1 (line 77), TCR stimulation (line 125) or DC immunization (line 137).

Accordingly, we now define programmed cell death protein 1 (PD-1) on line 36, programmed death-ligand 1 (PD-L1) on line 79, T-cell receptor (TCR) on line 136, and dendritic cell (DC) on line 67.

Line 115-116: please correct “we did not observe significant differences in expression levels of (...), *Col3a1*, ...”. According to Figure 2b, expression of *Col3a1* is significantly decreased in EAM+ICI compared to EAM.

Note that in Figure 2b, we show that *Col3a1* expression levels are comparable between EAM and EAM + ICI models, while those of *Col1a1* significantly decrease in the EAM+ICI compared to the EAM model. We now state this in the Results on Page 7, Line 124–127.

Line 146: Correct “chronic heart failure” by chronic heart dysfunction. There is no evidence of HF.

Accordingly, we revised this wording in the revision.

Line 182: “one reported ANGPTL2 receptor is integrin $\alpha 5\beta 1$ ”; since all the references quoted are from this team, the sentence could be changed to “we reported”.

We have corrected this wording in the revision.

REVIEWERS' COMMENTS:

Reviewer #1 (Remarks to the Author):

ANGPTL2 promotes immune checkpoint inhibitor-1 related autoimmune myocarditis by Horiguchi et al (Manuscript ID: COMMSBIO-23-1294A; Communications Biology).

The authors have addressed all my comments for this paper. The paper has been significantly improved after revising.

Reviewer #2 (Remarks to the Author):

The revision of the manuscript does take into account and answers my concerns. I appreciate the precisions made to the model methods part, animal number, technical details... The discussion has been improved.

Reviewer #3 (Remarks to the Author):

No more comments, great study.